# It Is Likely That Your Loss Should be a Likelihood

**Mark Hamilton**
MIT, Microsoft
markth@mit.edu

**Evan Shelhamer**
MIT, Adobe Research
shelhamer@adobe.com

**William Freeman**
MIT, Google
billf@mit.edu

## Abstract

Many common loss functions such as mean-squared-error, cross-entropy, and reconstruction loss are unnecessarily rigid. Under a probabilistic interpretation, these common losses correspond to distributions with fixed shapes and scales. We instead argue for optimizing full likelihoods that include parameters like the normal variance and softmax temperature. Joint optimization of these "likelihood parameters" with model parameters can adaptively tune the scales and shapes of losses in addition to the strength of regularization. We explore and systematically evaluate how to parameterize and apply likelihood parameters for robust modeling, outlier-detection, and re-calibration. Additionally, we propose adaptively tuning $L_2$ and $L_1$ weights by fitting the scale parameters of normal and Laplace priors and introduce more flexible element-wise regularizers.

## 1 Introduction

Choosing the right loss matters. Many common losses arise from likelihoods, such as the squared error loss from the normal distribution , absolute error from the Laplace distribution, and the cross entropy loss from the softmax distribution. The same is true of regularizers, where $L_2$ arises from a normal prior and $L_1$ from a Laplace prior.

Deriving losses from likelihoods recasts the problem as a choice of distribution which allows data-dependent adaptation. Standard losses and regularizers implicitly fix key distribution parameters, limiting flexibility. For instance, the squared error corresponds to fixing the normal variance at a constant. The full normal likelihood retains its scale parameter and allows optimization over a parametrized set of distributions. This work examines how to jointly optimize distribution and model parameters to select losses and regularizers that encourage generalization, calibration, and robustness to outliers. We explore three key likelihoods: the normal, softmax, and the robust regression likelihood $\rho$ of Barron (2019). Additionally, we cast adaptive *priors* in the same light and introduce adaptive regularizers. Our contributions:

1. We systematically survey and evaluate global, data, and predicted likelihood parameters and introduce a new self-tuning variant of the robust adaptive loss $\rho$
2. We apply likelihood parameters to create new classes of robust models, outlier detectors, and re-calibrators.
3. We propose adaptive versions of $L1$ and $L2$ regularization using parameterized normal and Laplace priors on model parameters.

## 2 Background

**Notation** We consider a dataset $\mathcal{D}$ of points $x_i$ and targets $y_i$ indexed by $i \in \{1, \dots, N\}$. Targets for regression are real numbers and targets for classification are one-hot vectors. The model $f$ with parameters $\theta$ makes predictions $\hat{y}_i = f_\theta(x)$. A loss $L(\hat{y}, y)$ measures the quality of the prediction given the target. To learn model parameters we solve the following loss optimization:

$$\min_\theta \mathop{\mathbb{E}}_{(x,y)\sim\mathcal{D}} L(\hat{y} = f_\theta(x), y) \qquad (1)$$

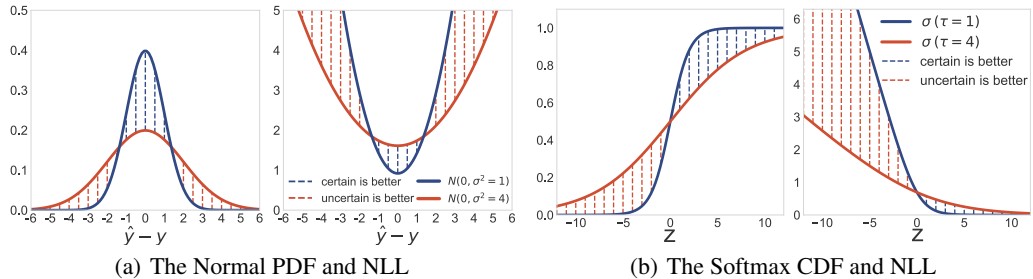

(a) The Normal PDF and NLL          (b) The Softmax CDF and NLL

Figure 1: Optimizing likelihood parameters adapts the loss without manual hyperparameter tuning to balance accuracy and certainty.

A likelihood $\mathcal{L}(\hat{y}|y,\phi)$ measures the quality of the prediction as a distribution over $\hat{y}$ given the target $y$ and likelihood parameters $\phi$. We use the negative log-likelihood $\ell$ (NLL), and the likelihood interchangeably since both have the same optima. We define the full likelihood optimization:

$$\min_{\theta,\phi} \; \mathbb{E}_{(x,y)\sim\mathcal{D}} \; \ell(\hat{y}=f_\theta(x)|y,\phi) \tag{2}$$

to jointly learn model and likelihood parameters. "Full" indicates the inclusion of $\phi$, which controls the distribution and induced NLL loss. We focus on full likelihood optimization in this work. We note that the target, $y$, is the only supervision needed to optimize model and likelihood parameters, $\theta$ and $\phi$ respectively. Additionally, though the shape and scale varies with $\phi$, reducing the error $\hat{y}-y$ always reduces the NLL for our distributions.

**Distributions Under Investigation** This work considers the normal likelihood with variance $\sigma$ (Bishop et al., 2006; Hastie et al., 2009), the softmax likelihood with temperature $\tau$ (Hinton et al., 2015), and the robust likelihood $\rho$ (Barron, 2019) with shape $\alpha$ and scale $\sigma$ that control the scale and shape of the likelihood. ==The first two are among the most common losses in machine learning, and the last loss provides an important illustration of a likelihood parameter that affects "shape" instead of "scale".== We note that changing the scale and shape of the likelihood distribution is not "cheating" as there is a trade-off between uncertainty and credit. Figure 1 shows how this trade-off affects the Normal and softmax distributions and their NLLs.

The normal likelihood has terms for the residual $\hat{y}-y$ and the variance $\sigma$ as

$$\mathcal{N}(\hat{y}|y,\sigma) = (2\pi\sigma^2)^{-\frac{1}{2}} \exp\left(-\frac{1}{2}\frac{(\hat{y}-y)^2}{\sigma^2}\right), \tag{3}$$

with $\sigma \in (0,\infty)$ scaling the distribution. The normal NLL can be written $\ell_{\mathcal{N}} = \frac{1}{2\sigma^2}(\hat{y}-y)^2 + \log\sigma$, after simplifying and omitting constants that do not affect minimization. We recover the squared error by substituting $\sigma = 1$.

The softmax defines a categorical distribution defined by scores $z$ for each class $c$ as

$$\text{softmax}(\hat{y}=y|z,\tau) = \frac{e^{z_y\tau}}{\sum_c e^{z_c\tau}}, \tag{4}$$

with the temperature, $\tau \in (0,\infty)$, adjusting the entropy of the distribution. We recover the classification cross-entropy loss, $-\log p(\hat{y}=y)$, by substituting $\tau = 1$ in the respective NLL. We state the gradients of these likelihoods with respect to their $\sigma$ and $\tau$ in Section A of the supplement.

The robust loss $\rho$ and its likelihood are

$$\rho(x,\alpha,\sigma) = \frac{|\alpha-2|}{\alpha}\left(\left(\frac{(x/\sigma)^2}{|\alpha-2|}+1\right)^{\alpha/2}-1\right) \text{ and} \tag{5}$$

$$p(\hat{y} \mid y,\alpha,\sigma) = \frac{1}{\sigma Z(\alpha)}\exp\left(-\rho(\hat{y}-y,\alpha,\sigma)\right), \tag{6}$$

with shape $\alpha \in [0, \infty)$, scale $\sigma \in (0, \infty)$, and normalization function $Z(\alpha)$. This robust loss, $\rho$, has the interesting property that it generalizes several different loss functions commonly used in robust learning such as the L2 loss ($\alpha = 2$), pseudo-huber loss (Charbonnier et al., 1997)($\alpha = 1$), Cauchy loss (Li et al., 2018) ($\alpha = 0$), Geman-McClure loss (Ganan & McClure, 1985), ($\alpha = -2$), and Welsch (Dennis Jr & Welsch, 1978) loss ($alpha = -\infty$). Learning the shape parameter allows models to adapt the shape of their noise distribution.

## 3  RELATED WORK

Likelihood optimization follows from maximum likelihood estimation (Hastie et al., 2009; Bishop et al., 2006), yet is uncommon in practice for fitting deep regressors and classifiers for discriminative tasks. However Kendall & Gal (2017); Kendall et al. (2018); Barron (2019); Saxena et al. (2019) optimize likelihood parameters to their advantage yet differ in their tasks, likelihoods, and parameterizations. In this work we aim to systematically experiment, clarify usage, and encourage their wider adoption.

Early work on regressing means and variances (Nix & Weigend, 1994) had the key insight that optimizing the full likelihood can fit these parameters and adapt the loss. Some recent works use likelihoods for loss adaptation, and interpret their parameters as the uncertainty (Kendall & Gal, 2017; Kendall et al., 2018), robustness (Kendall & Gal, 2017; Barron, 2019; Saxena et al., 2019), and curricula (Saxena et al., 2019) of losses. MacKay & Mac Kay (2003) uses Bayesian evidence to select hyper-parameters and losses based on proper likelihood normalization. Barron (2019) define a generalized robust regression loss, $\rho$, to jointly optimize the type and degree of robustness with global, data-independent, parameters. Kendall & Gal (2017) predict variances for regression and classification to handle data-dependent uncertainty. Kendall et al. (2018) balance multi-task loss weights by optimizing variances for regression and temperatures for classification. These global parameters depend on the task but not the data, and are interpreted as inherent task uncertainty. Saxena et al. (2019) define a differentiable curriculum for classification by assigning each training point its own temperature. These data parameters depend on the index of the data but not its value. We compare these different likelihood parameterizations across tasks and distributions.

In the calibration literature, Guo et al. (2017) have found that deep networks are often miscalibrated, but they can be re-calibrated by cross-validating the temperature of the softmax. In this work we explore several generalizations of this concept. Alternatively, Platt scaling (Platt, 1999) fits a sigmoid regressor to model predictions to calibrate probabilities. Kuleshov et al. (2018) re-calibrate regressors by fitting an Isotonic regressor to the empirical cumulative distribution function.

## 4  LIKELIHOOD PARAMETER TYPES

We explore the space of likelihood parameter representations for model optimization and inference. Though we note that some losses, like adversarial losses, are difficult to represent as likelihoods, many different losses in the community have a natural probabilistic interpretation. Often, these probabilistic interpretations can be parametrized in a variety of ways. We explore two key axes of generality when building these loss functions: conditioning and dimensionality.

**Conditioning**  We represent the likelihood parameters by three functional classes: global, data, and predicted. *Global* parameters, $\phi = c$, are independent of the data and model and define the same likelihood distribution for all points. *Data* parameters, $\phi_i$, are conditioned on the index, $i$, of the data, $x_i$, but not its value. Every training point is assigned an independent likelihood parameter, $\phi_i$ that define different likelihoods for each training point. *Predicted* parameters, $\phi(x) = g_\eta(x)$, are determined by a model, $g$, with parameters $\eta$ (not to be confused with the task model parameters $\theta$). Global and predicted parameters can be used during training and testing, but data parameters are only assigned to each training point and are undefined for testing. We show a simple example of predicted temperature in Figure 4, and an illustration of the parameter types in Figure 2.

We note that for certain global parameters like a learned Normal scale, changing the scale does not affect the optima, but does change the probabilistic interpretation. This invariance has led many authors to drop the scale from their formulations. However, when models can predict these scale parameters they can naturally remain calibrated in the presence of heteroskedasticity and outliers.

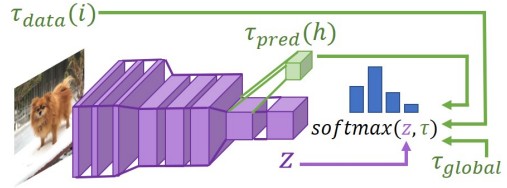

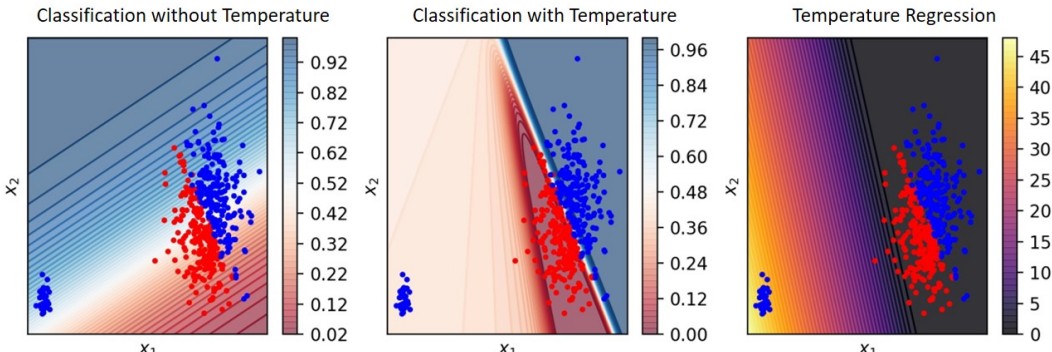

Figure 2: Illustration of an image classifier with three different types of likelihood temperature conditioning: global, predicted, and data. Each represents a different way to parametrize the model's temperature.

Figure 3: An image loss function with three different likelihood parameter dimensionalities. Each represents a possible way to parametrize the additional scale parameter added to the loss.

Figure 4: A synthetic logistic regression experiment. Regressing softmax temperature reduces the influence of outliers (blue, bottom-left), by locally raising temperature. The jointly optimized model (center and right panel) achieves a more accurate classification that a model trained without adaptive temperature (left panel).

Additionally we note that for the shape parameter of the robust likelihood, $\rho$, changing global parameters does affect model fitting. Previous works have adapted a global softmax temperature for model distillation (Hinton et al., 2015), and recalibration (Guo et al., 2017). Barron (2019) also experiments with global values of loss function shape and scale parameters. The main work on Data parameters is that of Saxena et al. (2019) who use these to learn a curriculum. Model-based parameters appear in earlier work on regressing variance (Nix & Weigend, 1994), and more recent work by Kendall & Gal (2017).

**Dimensionality** The dimensionality, $|\phi|$, of likelihood parameters can vary with the dimension of the task prediction, $\hat{y}$. For example, image regressors can use a single likelihood parameter for each image $|\phi| = 1$, RGB image channel $|\phi| = C$, or even every pixel $|\phi| = W \times H \times C$ as in Figure 3. These choices correspond to different likelihood distribution classes. Dimensionality and Conditioning of likelihood parameters can interact. For example, data parameters with $|\phi| = W \times H \times C$ would result in $N \times W \times H \times C$ additional parameters, where $N$ is the size of the dataset. This can complicate implementations and slow down optimization due to disk I/O when their size exceeds memory. Table 5 in the appendix contrasts the computational requirements of different likelihood parameter types. The work of Barron (2019) explores both scalar and pixel-wise dimensionalities for his robust loss.

## 5 APPLICATIONS

Table 1: MSE, Time, and Memory increase (compared to standard normal likelihood) for reconstruction by variational auto-encoders with different parameterizations of the robust loss, $\rho$. Predicted likelihood parameters yield more accurate reconstruction models.

| Param. | Dim | MSE | Time | Mem |
|--------|-----|-----|------|-----|
| Global | $1{\times}1{\times}1$ | 225.8 | **1.04**$\times$ | $<1$**KB** |
| Data | $1{\times}1{\times}1$ | 244.2 | 2.70$\times$ | 0.6GB |
| Pred. | $1{\times}1{\times}1$ | 228.5 | **1.04**$\times$ | $<1$MB |
| Global | $H{\times}W{\times}C$ | 231.1 | 1.08$\times$ | $<1$MB |
| Data | $H{\times}W{\times}C$ | 252.6 | 9.42$\times$ | 4.4GB |
| Pred. | $H{\times}W{\times}C$ | **222.3** | 1.08$\times$ | $<1$MB |

## 5.1 ROBUSTNESS AND OUTLIER DETECTION

Data in the wild is noisy, and machine learning methods should be robust to noise, heteroskedasticity, and corruption. Unfortunately, models trained with the standard mean squared error (MSE) loss are highly susceptible to outliers, and cannot naturally handle heteroskedasticity due to this loss' fixed variance (Huber, 2004). Allowing models to predict and optimize their likelihood parameters allows models to generalize to these more complex settings. More specifically, likelihood parameters naturally transform standard methods such as regressors, classifiers, and manifold learners into robust variants without expensive outer-loop of model fitting such as RANSAC (Fischler & Bolles, 1981) and Theil-Sen (Theil, 1992). Figure 4 demonstrates this effect with a simple classification dataset, and we point readers to Figures 9 of the Supplement for similar examples for regression and manifold learning.

In certain datasets, even the assumption of Gaussianity is too restrictive and one must consider more robust and long-tailed distributions. This has led many to investigate broader classes of likelihoods such as Generalized Linear Models (GLMs) (Nelder & Wedderburn, 1972) or the more recent general robust loss, $\rho$, of (Barron, 2019). To systematically explore how likelihood parameter dimension and conditioning affect model robustness and quality, we reproduce Barron (2019)'s variational auto-encoding (Kingma & Ba, 2015) (VAE) experiments on faces from the CelebA dataset (Liu et al., 2015) in Table 1. We explore learned data (Saxena et al., 2019) and model parameters in addition to Barron's learned global parameters. We also include two natural parameter dimensionalities: a single set of parameters for the whole image, and a set of parameters for each pixel and channel. We find that predicted parameters achieve the best performance while maintaining fast training time and a small memory footprint. We also find that pixel-wise learned parameters correlate with challenging areas of images and we visualize these parameters in Section D of the Appendix.

This experiment uses a $1 \times 1$ convolution on the last hidden layer of the decoder as a likelihood parameter model and has the same resolution as the output. The low and high dimensional losses use the same convolutional regressor, but the 1 dimensional case averages over pixels. In the high dimensional case, the output has three channels (for RGB), with six channels total for shape and scale regression. We use the same non-linearities to constrain the shape and scale outputs to reasonable ranges as in (Barron, 2019). More specifically, we use an affine sigmoid to keep the shape $\alpha \in [0, 3]$ and the softplus to keep scale $c \in [10^{-8}, \infty)$. Table 1 gives the results of evaluating each method by MSE on the validation set, while training each method with their respective loss parameters. Data parameter optimization uses Tensorflow's implementation of sparse RMSProp (Tieleman & Hinton, 2012). We also inherit weight decay $\|\phi\|_2^2$, gradient clipping $\nabla_\phi / \|\nabla_\phi\|_2^2$, and learning rate scaling $\alpha_\phi = \alpha \cdot m$ for learning rate $\alpha$ and multiplier $m$ from Barron (2019).

The robustness we see in our VAE experiments stems from the fact that likelihood parameter prediction gives models a direct channel to express their "uncertainty" for each data-point with respect to the task. This allows models to naturally down-weight and clean outliers from the dataset which can improve model robustness. Consequently, one can harness this effect to create outlier detectors from *any* underlying model architecture by using learned scales or temperatures as an outlier score function. Furthermore, predicted likelihood parameters allow these methods to detect outliers in unseen data. In Figure 5 we show how auditing temperature or noise parameters can help practitioners spot erroneous labels and poor quality examples. In particular, the model-parameterized temperatures of

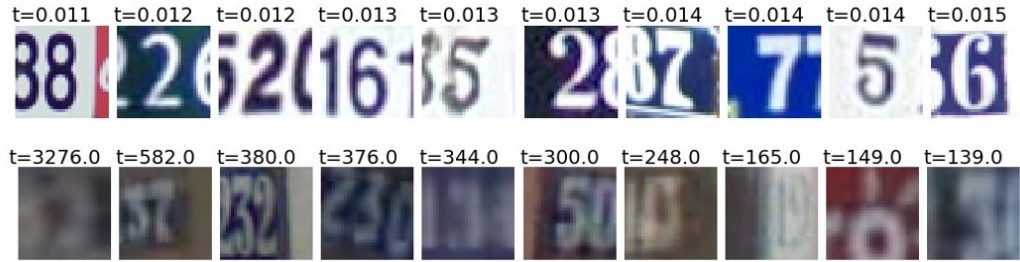

Figure 5: The data with the lowest (top) and highest (bottom) predicted temperatures in the SVHN dataset. High temperature entries are blurry, cropped poorly, and generally difficult to classify.

Table 2: Median outlier detection performance of several methods across 22 benchmark datasets from ODDS.

| Method | Median AUC |
|---|---|
| LOF | .669 |
| FB | .702 |
| ABOD | .727 |
| AE | .737 |
| VAE | .792 |
| COPOD | .799 |
| PCA | .808 |
| OCSVM | .814 |
| MCD | .820 |
| KNN | .822 |
| HBOS | .822 |
| IF | .823 |
| CBLOF | .836 |
| AE+S (Ours) | .846 |
| **PCA+S (Ours)** | **.868** |

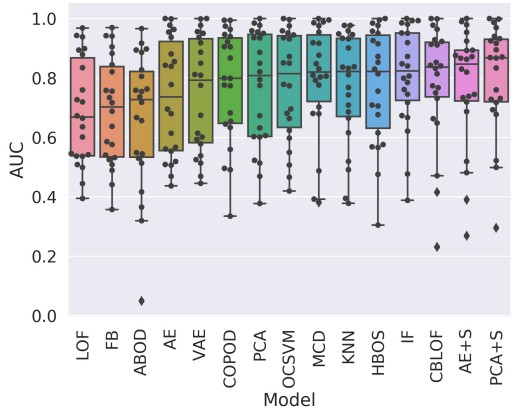

Figure 6: Distribution of Outlier Detection AUC across the ODDS Benchmark. Our approaches, PCA+S and AE+S, are competitive with other Outlier Detection systems.

an image classifier (trained using the setup of 5.3) correlates strongly with blurry, dark, and difficult examples on the Street View House Number (SVHN) dataset. We use this approach to create simple outlier detection algorithms by considering deep (AE+S) and linear (PCA+S) auto-encoders (Kramer, 1991) with data-conditioned scale parameters as outlier scores. We evaluate this approach on tabular datasets using deep and linear auto-encoders with model-parameterized scales. In Table 2 we quantitatively demonstrate the quality of these simple likelihood parameter approaches across 22 datasets from the Outlier Detection Datasets (ODDS), a standard outlier detection benchmark (Rayana, 2016). The ODDS benchmark supplies ground truth outlier labels for each dataset, which allows one to treat outlier detection as an unsupervised classification problem. We compare against a variety of established outlier detection approaches included in the pyOD (Zhao et al., 2019) framework including: One-Class SVMs (OCSVM) (Schölkopf et al., 2000), Local Outlier Fraction (LOF) (Breunig et al., 2000), Angle Based Outlier Detection (ABOD) (Kriegel et al., 2008), Feature Bagging (FB) (Lazarevic & Kumar, 2005), Auto Encoder Distance (AE) (Aggarwal, 2015), K-Nearest Neighbors (KNN) (Ramaswamy et al., 2000; Angiulli & Pizzuti, 2002), Copula Based Outlier Detection (COPOD) (Li et al., 2020), Variational AutoEncoders (VAE) (Kingma & Welling, 2013), Minimum Covariance Determinants with Mahlanohbis Distance (MCD) (Rousseeuw & Driessen, 1999; Hardin & Rocke, 2004), Histogram-based Outlier Scores (HBOS) (Goldstein & Dengel, 2012), Principal Component Analysis (PCA) (Shyu et al., 2003), Isolation Forests (IF) (Liu et al., 2008; 2012), and the Clustering-Based Local Outlier Factor (CBLOF) (He et al., 2003).

Our predicted scale auto-encoders use PyTorch's layers API (Paszke et al., 2019) with rectified linear unit (ReLU) activations for deep auto-encoders and Glorot uniform initialization (Dahl et al., 2013; Glorot & Bengio, 2010) for all layers. We use Adam (Kingma & Ba, 2015) with a learning rate of

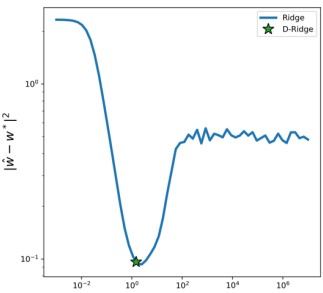 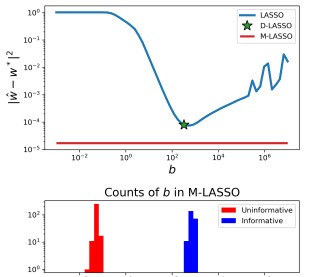 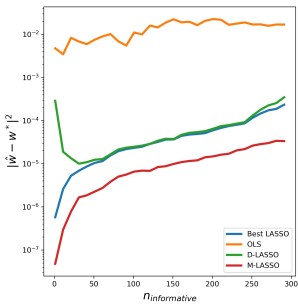

Figure 7: Performance of $L2$ (left) and $L1$ (middle) regularized linear regression on a 500 dimensional synthetic dataset where the true parameters, $w^*$, are known. Dynamic Ridge (D-Ridge) and D-LASSO regression find the regularization strength that best estimates the true parameters. M-LASSO outperforms any single global regularization strength and does not shrink informative weights. (right) Performance of adaptive $L1$ regularization methods as a function of true model sparsity. In all cases, Multi-LASSO outperforms other methods by orders of magnitude.

.0005 for 4000 steps with 20% dropout before the code space. We follow ODDS guidelines and standard scale the data prior to fitting.

Our methods (PCA+S and AE+S) use a similar principle as isolation-based approaches that determine outliers based on how difficult they are to model. In existing approaches, outliers influence and skew the isolation model which causes the model to exhibit less confidence on the whole. This hurts a model's ability to distinguish between inliers and outliers. In contrast, our approach allows the underlying model to down-weight outliers. This yields a more consistent model with a clearer decision boundary between outliers and inliers as shown in Figure 4. As a future direction of investigation we note that our approach is model-architecture agnostic, and can be combined with domain-specific architectures to create outlier detection methods tailored to images, text, and audio.

## 5.2 ADAPTIVE REGULARIZATION WITH PRIOR PARAMETERS

In addition to optimizing the shape and scale of the likelihood distribution of the model output, we can use the same approach to optimize the *prior* distribution of the model parameters. More specifically, we propose adaptive regularizers for a model's parameters, $\theta$. This approach optimizes the distribution parameters of the prior, $\phi_{\text{prior}}$, to naturally tune the degree of regularization. In particular, the Normal (Ridge, L2) and Laplace (LASSO, L1) priors, with scale parameters $\sigma$ and $b$, regularize model parameters for small magnitude and sparsity respectively (Hastie et al., 2009). The degree of regularization, $\lambda \in [0, \infty)$, is conventionally a hyperparameter of the regularized loss function:

$$\min_{\theta} \sum_i^N (\hat{y}_i := f_\theta(x_i) - y_i)^2 + \lambda \sum_j^P |\theta_j|. \tag{7}$$

We note that we cannot choose $\lambda$ by direct minimization because it admits a trivial minimum at $\lambda = 0$. In the linear case, one can select this weight efficiently using Least Angle Regression (Efron et al., 2004). However, in general $\lambda$ is usually learned through expensive cross validation methods. Instead, we retain the prior with its scale parameter, and jointly optimize over the full likelihood:

$$\min_{\theta, \sigma, b} \sum_i^N \left( \frac{1}{2\sigma^2} (\hat{y}_i - y_i)^2 + \log \sigma \right) + \sum_j^P \left( \frac{|\theta_j|}{b} + \log b \right) \tag{8}$$

This approach, the Dynamic Lasso (D-LASSO), admits no trivial solution for the prior parameter $b$, and must balance the effective regularization strength, $\frac{1}{b}$, with the normalization factor, $\log b$. D-LASSO selects the degree of regularization by gradient descent, rather than expensive black-box search. In Figure 7 (left) and (middle) we show that this approach, and its Ridge equivalent, yield ideal settings of the regularization strength on a suite of synthetic regression problems. Figure 7

(right) shows D-LASSO converges to the best LASSO regularization strength for a variety of true-model sparsities. As a further extension, we replace the global $\sigma$ or $b$ with a $\sigma_j$ or $b_j$ for each model parameter, $\theta_j$, to locally adapt regularization to each model weight (**M**ulti-Lasso). This consistently outperforms any global setting of the regularization strength and shields important weights from undue shrinkage 7 (middle). For our experiments we use 500 samples of 500 dimensional normal distributions mapped through linear functions with additive gaussian noise. Linear transformations use Uniform$[1, 2]$ weights and LASSO experiments use sparse transformations. We use tensorflow's Adam optimizer with $lr = .0005$ for 100000 steps.

Our approach of learning regularizer scale parameters can be viewed naturally through the lens of hierarchical priors (Gelman et al., 2013). More specifically this approach is implicitly performing maximum a posteriori (MAP) inference on the prior's scale with respect to a uniform prior on that parameter. We note that though these methods for hyperparameter selection are common in the Bayesian literature, they are not widely used in practice in the deep learning community. This work aims to bring these parameters back within the scope of deep learning where they can be easily expanded to more flexible forms such as our introduced Multi-Lasso.

## 5.3   RE-CALIBRATION

The work of (Guo et al., 2017) shows that modern networks are accurate, yet systematically over-confident, a phenomenon called mis-calibration. We investigate the role of optimizing likelihood parameters to re-calibrate models. More specifically, we can fit likelihood parameter regressors on a validation set to modify an existing model's confidence to better align with the validation set. This approach is a generalization of Guo et al. (2017)'s Temperature Scaling method, which we refer to as Global Scaling (GS) for notational consistency. Global Scaling re-calibrates classifiers with a learned global parameter, $\tau$ in the loss function: $\sigma(\vec{x}, \tau)$.

Fitting model-conditioned likelihood parameters to a validation set defines a broad class of re-calibration strategies. From these we introduce three new re-calibration methods. Linear Scaling (LS) learns a linear mapping, $l$, to transform logits to a softmax temperature: $\sigma(\vec{x}, l(\vec{x}))$. Linear Feature Scaling (LFS) learns a linear mapping, $l$, to transform the *features* prior to the logits, $\vec{f}$, to a softmax temperature: $\sigma(\vec{x}, l(\vec{f}))$. Finally, we introduce Deep Scaling (DS) for regressors which learns a nonlinear network, $N$, to transform features, $\vec{f}$, into a temperature: $\sigma(\vec{x}, N(\vec{f}))$.

In Table 3 we compare our recalibration approaches to the previous state of the art: Global Scaling. We note that (Guo et al., 2017) have already shown that Global Scaling outperform Bayesian Binning into Quantiles (Naeini et al., 2015), Histogram binning (Zadrozny & Elkan, 2001), and Isotonic Regression. We recalibrate both ResNet50 (He et al., 2016) and DenseNet121 (Huang et al., 2017) on a variety of vision datasets. We measure classifier miscalibration using the Expected Calibration Error (ECE) (Guo et al., 2017) to align with prior art. We additionally evaluate Isotonic recalibration, Platt Scaling (Platt, 1999), and Vector Scaling (VS) (Guo et al., 2017), which learns a vector, $\vec{v}$, to re-weight logits: $\sigma(\vec{v}\vec{x}, 1)$. LS and LFS tend to outperform other approaches like GS and VS, which demonstrates that richer likelihood parametrizations can improve calibration akin to how richer models can improve prediction.

Our experiments leverage Tensorflow's Dataset APIs that include the SVHN, (Netzer et al., 2011), ImageNet (Deng et al., 2009), CIFAR-100, CIFAR-10 (Krizhevsky, 2009) datasets. We use Keras implementations of DenseNet-121 (Huang et al., 2017) and ResNet-50 (He et al., 2016) with default initializations. For optimization we use Adam with $lr = 0.0001, \beta_1 = .9, \beta_2 = .99$ (Kingma & Ba, 2015) and train for 300 epoch with a batch size of 512.

For recalibrating regressors, we compare against the previous state of the art, Kuleshov et al. (2018), who use an Isotonic regressor to correct a regressors' confidence. We use the same experimental setting as Kuleshov et al. (2018) including the UCI datasets (Dua & Graff, 2017), and regressor calibration metric (CAL). Table 4 shows that our approaches can outperform this baseline as well as the regression equivalent of Global Scaling. Inputs and targets are scaled to unit norm and variance prior to fitting for all regression experiments and missing values are imputed using scikit-learn's "SimpleImputer" (Pedregosa et al., 2011). Experiments utilize Keras' layers API with two hidden rectified linear unit (ReLU) layers, Glorot uniform initialization (Dahl et al., 2013; Glorot & Bengio, 2010) and Adam optimization with $lr = 0.001$ for 3000 steps without minibatching.

Table 3: Comparison of calibration methods by ECE for ResNet-50 (RN50) and DenseNet-121 (DN121) architectures on test data. Our predicted likelihood parameter methods: Linear Scaling (LS) and Linear Feature Scaling (LFS) outperform other approaches. In all cases our methods reduce miscalibration with comparable computation time as GS.

| Model | Dataset | Uncalibrated | Platt | Isotonic | GS | VS | LS | LFS |
|-------|---------|--------------|-------|----------|------|------|--------|--------|
| RN50 | CIFAR-10 | .250 | .034 | .053 | .046 | .037 | **.018** | **.018** |
| RN50 | CIFAR-100 | .642 | .061 | .072 | .035 | .044 | **.030** | .173 |
| RN50 | SVHN | .072 | .053 | .010 | .029 | .022 | **.009** | **.009** |
| RN50 | ImageNet | .430 | .018 | .070 | .019 | .023 | .026 | **.015** |
| DN121 | CIFAR-10 | .253 | .048 | .042 | .039 | .034 | **.028** | **.028** |
| DN121 | CIFAR-100 | .537 | .049 | .067 | .024 | .024 | **.014** | .031 |
| DN121 | SVHN | .079 | .018 | **.010** | .022 | .017 | .011 | **.010** |
| DN121 | ImageNet | .229 | .028 | .095 | .021 | **.019** | .043 | **.019** |

Table 4: Comparison of regression calibration methods as evaluated by their calibration error as defined in (Kuleshov et al., 2018). Predicted likelihood parameters often outperform other methods.

| Dataset | Uncalibrated | Isotonic | GS | LS | DS |
|---------|--------------|----------|--------|------------|------------|
| crime | 0.3624 | 0.3499 | 0.0693 | **0.0125** | 0.0310 |
| kinematics | 0.0164 | 0.0103 | 0.0022 | **0.0021** | 0.0032 |
| bank | 0.0122 | 0.0056 | 0.0027 | 0.0024 | **0.0020** |
| wine | 0.0091 | 0.0108 | 0.0152 | 0.0131 | **0.0064** |
| mpg | 0.2153 | 0.2200 | 0.1964 | 0.1483 | **0.0233** |
| cpu | 0.0862 | **0.0340** | 0.3018 | 0.2078 | 0.1740 |
| soil | 0.3083 | **0.3000** | 0.3130 | 0.3175 | 0.3137 |
| fried | 0.0006 | **0.0002** | **0.0002** | **0.0002** | **0.0002** |

## 6 EXPERIMENTAL DETAILS

We run all experiments on Ubuntu 16.04 Azure Standard NV24 virtual machines (24 CPUs, 224 Gb memory, and $4\times$ M60 GPUs) with Tensorflow 1.15 (Abadi et al., 2015) and PyTorch 1.17 (Paszke et al., 2019). Many likelihood parameters have constrained domains, such as the normal variance $\sigma \in [0, \infty)$. To evade the complexity of constrained optimization, we define unconstrained parameters $\phi_u$ and choose a transformation $t(\cdot)$ with inverse $t^{-1}(\cdot)$ to map to and from the constrained $\phi$. For positivity, $\exp/\log$ parameterization is standard (Kendall & Gal, 2017; Kendall et al., 2018; Saxena et al., 2019). However, this parameterization can lead to instabilities and we use the softplus, $s^+(x) = \log(1 + \exp(x))$, instead. Shifting the softplus, $s_c^+(x) = (ln(1 + e^x) + c)/(ln(2) + c)$, further improves stability and we explore this effect in Figure 12 of the Appendix. We use an affine softplus $s_{.01}^+$ and $s_{.2}^+$ respectively for adaptive scales and temperatures respectively. The one exception is adaptive regularizer scales where we found $\exp$ led to faster convergence. For the constrained interval $[a, b]$ we use affine transformations of the sigmoid $s(x) = \frac{1}{1+\exp(-x)}$ (Barron, 2019). We initialize likelihood parameter biases to settings that yield MSE and Cross Entropy ($\sigma = \tau = 1$).

## 7 CONCLUSION

Optimizing the full likelihood can improve model quality by adapting losses and regularizers. Full likelihoods are agnostic to the architecture, optimizer, and task, which makes them simple substitutes for standard losses. Global, data, and predicted likelihood parameters offer different degrees of expressivity and efficiency. In particular, predicted parameters adapt the likelihood to each data point during training and testing without significant time and space overhead. By including these parameters in a loss function one can improve a model's robustness and generalization ability and create new classes of outlier detectors and recalibrators that outperform baselines. More generally, we hope this work encourages joint optimization of model and likelihood parameters, and argue it is likely that your loss should be a likelihood.

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
