# OpenReview forum: "It Is Likely That Your Loss Should be a Likelihood"
_ICLR.cc/2021/Conference — Reject_

### Official Review · AnonReviewer4 · 2020-10-28
**An interesting idea but writing and presentation should be improved.**

**Rating:** 6
**Confidence:** 4

**Review:**

# Summary:
The paper proposes the use of complete parametrized likelihoods for providing supervision in place of the commonly used loss functions. The normal distribution, the categorical distribution defined by softmax and the likelihood of the robust rho-estimator are considered. The main idea is that by including the parameters of these likelihoods and optimizing over them, one can increase robustness, detect outliers and achieve re-calibration. In addition, by considering parametric priors and tuning their parameters one can obtain more flexible regularizers over the trainable parameters of a model.

# Strengths:
The idea of the paper is quite interesting as it lifts some commonly made and often overlooked assumptions regarding the data distribution. By lifting these assumptions one can improve the performance of the trained model by considering likelihoods that better capture the data distribution. For example, data is usually affected by heteroskedasticity as well as outliers, and if the likelihood considered in its full form covers these aspects the resulting models will be better calibrated.
The proposed methods consider different aspects of conditioning and dimensionality of the likelihoods employed, varying from global to data-specific modeling.

The use of likelihoods instead of common loss functions leads to competitive new methods and variants for robust modeling, outlier detection, adaptive regularization and model re-calibration.


# Weaknesses:
Although the use of likelihoods instead of loss functions is not a common practice in deep learning, its advantages have been thoroughly studied in statistics, econometrics and other disciplines, as also discussed in the related work of the paper. Hence, the novelty mainly lies in the application of these ideas in deep learning and the employment of some likelihoods better suited for the respective problems (i.e. softmax and rho-estimators).

The paper is interesting however I found it somewhat difficult to read. In my view it tries to pack many different aspects and applications of the main idea (use of likelihood) into a very limited space. In fact, there are too many cross-references to the supplemental material, to the point that it seems that most of the paper is described in the supplemental material.
On a similar note, due to the fact that four different application domains are considered, there numerous methods, metrics and datasets involved in each one of them which are not sufficiently covered in the text. Additionally, many of the proposed methods/improvements/variants on each domain are not explained in sufficient detail (e.g. AE+S and PCA+S in Sec. 5.2). I would expect some more principled and thorough guidance on how to use the likelihood functions and, regarding the conditioning and dimensionality, strategies on how to choose among the various options.

Also some editing is required, for example the likelihood of the softmax is not provided as the respective sentence after eq. 4 is suddenly interrupted (see also the comments below).

## Minor comments
* the text in the figure is very small, making it very difficult to read in typical zoom levels (~100%)
* Figure 3: the text does not correspond to the figure for the intermediate case
* Figure 4, caption: include reference to left, middle and right panel
* Table 1: there is no reference of this table in the text. Also, the three dots should be replaced with the actual setting.


# Rating Justification:
I think that the overall idea of the paper is interesting and provides improved data modeling which leads to important advantages of the estimated models. However, possibly due to space limitations, the paper does not explain in sufficient detail important aspect of applying the proposed idea in the domains considered.

# Rating and comments after the rebuttal
I think that in the revised version the paper has addressed many of the weaknesses pointed out in our reviews, hence I increase my rating to 6. Nevertheless, the paper still packs too much information which makes it difficult to read and appreciate.
Regarding novelty, although I agree with other reviews that the core idea is not novel I think that it is important that the paper stresses the applicability and usefulness of considering likelihoods in deep learning models, as it appears to be not fully appreciated currently.
Overall, I think that the paper would shine as a journal paper while it is only a borderline submission in its current form.

---

> ### Author Response · Authors · 2020-11-19
> **Simplifications, more experimental details, and addressing your comments**
>
> We thank the reviewer for their detailed and thoughtful feedback! Our point-by-point reply follows, and a revision has been uploaded. We tried to simplify the presentation and add extra details to help explain our experiments and design decisions. Regarding the formatting in our revision, light yellow areas represent re-worked sections and dark yellow areas are new additions and specific changes.
>
> >I found it somewhat difficult to read. In my view it tries to pack many different aspects and applications of the main idea (use of likelihood) into a very limited space.
>
> We appreciate the feedback to focus on fewer experiments and fuller details. To this end, we have removed the robust regression content and used the space to better cover outlier detection and add experimental details for all sections. We have likewise combined robust modelling and outlier detection into a single section. We unify these two because robustness and outlier detection are two sides of the same coin: for a model to be robust it must identify and “down-weight” outliers. In this light, we highlight CelebA VAE and ODDS outlier detection results, both of which apply likelihood parameters to unsupervised models.
>
> > many of the proposed methods/improvements/variants on each domain are not explained in sufficient detail (e.g. AE+S and PCA+S in Sec. 5.2)
>
> Good suggestion! We have used the extra page (for the rebuttal and camera-ready) and space we saved by removing some minor experiments to expand topics and include experimental details for all experiments. We also added an explicit experimental detail section for the whole work. We also made sure to include a paragraph to the outlier detection section to detail the experimental setup and how AE+S and PCA+S were built. We also re-worked the text to set this up better.
>
>
> > In fact, there are too many cross-references to the supplemental material, to the point that it seems that most of the paper is described in the supplemental material.
>
> We have incorporated the supplementary content in Sections I and J on experimental details into the text for easier reading.
>
> > I would expect some more principled and thorough guidance on how to use the likelihood functions and, regarding the conditioning and dimensionality, strategies on how to choose among the various options.
>
> We have sought to make this a bit clearer in our discussion of the results of the CelebA VAE experiment. In particular we find that model parametrized likelihoods offer a nice compromise between expressivity, speed, and simplicity.
>
> Regarding your “minor Comments”: Thank you for helping us spot these typos! We have fixed  all of these comments and increased the size of fonts in our figures.
>
> Again we thank you for the thoughtful feedback and hope that our changes can improve your outlook on the work. Please let us know if you think these changes address your feedback, or if you have any additional feedback that we can use to help improve the work.

---

### Official Review · AnonReviewer3 · 2020-10-29
**Training hyperparameters of the loss: experimental results**

**Rating:** 6
**Confidence:** 2

**Review:**

### Summary
The authors propose to test experimentally training of the loss hyperparameters. Given a loss $L$ with a reconstruction part $\ell$ and a penalty part $r$, they train the hyperparameters of $\ell$ and of $r$.

### Clarity
The main clarity issue is the reference formatting. Finding the cited papers in the references is tedious: hypertext links towards the *References* section should be added.

### Details
The main contribution is the experimental part. The authors have tested loss hyperparameters learning mainly in robustness against outliers and outlier detection.
I am not a specialist in outlier detection, so I cannot evaluate fairly most of the experimental results.

Table 2, CRIME dataset with MSE: it seems that "Base" works better than "Temp".

### Comments
From a Bayesian point of view, there is no problem with learning hyperparameters in the reconstruction part $\ell$ of the loss $L$. This is just equivalent to extending the family of probabilistic models. The user should only verify that $\ell$ can be interpreted as the negative log of a probability distribution, that is, $\exp(- \ell)$ is integrable.
However, training the hyperparameters of the penalty part of $L$ corresponds to training the parameters of the *prior* distribution, which is not acceptable (as such) from a Bayesian point of view.

Weak accept: optimization of hyperparameters of $\ell$ is theoretically well-founded from a Bayesian point of view, and should be more explored, as the authors do. However: 1) there is no consideration for this approach, while the preceding works of Barron (cited in the paper) had a word about it. This is important since we are still talking about a *likelihood*. 2) I am not sure whether the experimental results are significant enough. 3) The generalization of this approach to the penalty is not well-founded (at least, the authors do not justify it).

Edit:
### Rebuttal
I had read the rebuttal and the other reviews. It seems that there are some clarity issues, which are independent from my knowledge of the area chosen for the experiments. Moreover, outlier detection is not necessarily the only possible application.
However, I consider that the Bayesian point of view (which comes from preceding papers) has been well highlighted in the revised version.
These points put together, and given that I'm not sure of the significance of the experimental part, I do not change my rating.

---

> ### Author Response · Authors · 2020-11-19
> **Response**
>
>
> We thank the reviewer for their detailed and thoughtful feedback! In our revision, we tried to simplify the presentation and add extra details to help explain our experiments and design decisions. Regarding the formatting in our revision, light yellow areas represent re-worked sections and dark yellow areas are new additions and specific changes.
>
> We especially appreciate your feedback on the optimization of priors. In the section on optimizing priors, we explicitly connected this to MAP optimization on a Bayesian hierarchical prior with a uniform distribution. In this light optimization of priors is well-founded. We hope this speaks to your concerns and please let us know if you would like to see additional material on this topic.
>
> Regarding your comment on formatting our updated paper has links, but our previous paper did as well at the time of submission. We included a version of the paper in the supplemental which should retain its hyperlinks if the problem persists.
>
> Again we thank you for the thoughtful feedback and hope that our changes can improve your outlook on the work. Please let us know if you think these changes address your feedback, or if you have any additional feedback that we can use to help improve the work.

---

### Official Review · AnonReviewer1 · 2020-10-29
**This paper investigates loss functions from a likehood viewpoint and propose to optimize full likelihoods for learning purposes.**

**Rating:** 5
**Confidence:** 5

**Review:**

This paper studied loss functions by interpreting them from a likelihood viewpoint and by proposing to optimize "full" likelihoods for robust modeling, outlier-detection, and re-calibration purposes. Many loss functions stem from maximum likelihood estimation (MLE). For instance, the quadratic loss stems from MLE under Gaussianity, the absolute deviation loss stems from MLE under the Laplace noise assumption, and the check loss from MLE under the asymmetry Laplace noise assumption. What is common about these noise assumptions is that they contain scale/location parameters. Traditionally, in the machine learning community, these loss functions are used by ignoring the underlying noise assumptions and so the scale parameters are abandoned. The point of the paper is to take into account these scale parameters when solving the resulting optimization problems in learning.


Pros:
The problem studied in this paper is interesting. By investigating the full likelihood when applying loss functions, it does bring back our attention to the origin of loss functions. The three applications mentioned in this paper are also typical and important.

Cons:

1. My main concern is about the novelty of this paper. As mentioned above, I believe that it does bring back our attention to the origin of loss functions. However, given that the main point of this paper is to optimize full likelihoods for learning problems, it is not clear to me what are the real novel contributions made by the paper. I'm questioning this as simultaneous estimation scale/location parameters and target functions seems to be traditional approaches in parametric robust statistics. I'm expecting more comments in this regard.

2. The authors proposed to investigated three distributions, namely, Gaussian, softmax, and the distribution from Barron (2019). In my opinion, the three distributions are merely examples from three different scale families of distributions. I think more comments should be given when investigating the three specific ones.

3. Three types of parameters are mentioned, namely, global parameters, data parameters, and predicted parameters, which are used to categorize distributions. I would suggest that the authors should at least provide several examples by pointing out which is which.

4. In my opinion, the presentation of the paper could be further improved. For instance, the methodologies that the authors proposed become clear to me only after section 5.3. In addition, the title of this paper seems to be inappropriate in describing the main content of the paper.

---

> ### Author Response · Authors · 2020-11-19
> **Response**
>
> We thank the reviewer for their detailed and thoughtful feedback! In our revision we tried to simplify the presentation and add extra details to help explain our experiments and design decisions. Regarding the formatting in our revision, light yellow areas represent re-worked sections and dark yellow areas are new additions and specific changes.
>
> > I'm questioning this as simultaneous estimation scale/location parameters and target functions seems to be traditional approaches in parametric robust statistics. I'm expecting more comments in this regard.
>
> We have added some comments on how these approaches are inspired by the simultaneous regression of means and scales. One of the aspects that we think makes this work novel is the extension of this principle to deep models, and the exploration of its diverse applications which are often overlooked. One of the main points of the work is to show that these types of parameters are neglected in the deep learning community, and even the most simple applications of these help remedy many important issues such as robustness, heteroskedasticity, regularization tuning, and calibration, and do so with minimal overhead. Moreover these approaches are architecture agnostic, and can apply to a variety of different data types and architectures. Please let us know if this comes across in the reworked submission or if you would like us to expand on these points more.
>
> >In my opinion, the three distributions are merely examples from three different scale families of distributions. I think more comments should be given when investigating the three specific ones.
>
> Thank you for this feedback. We added a few comments on how the robust loss of Barron goes beyond just scale optimization and into shape optimization, ad the parameter $\alpha$ allows this loss to generalize several other losses in the literature such as the L2 loss (α = 2), pseudo-huber loss (Charbonnier et al., 1997)(α = 1), Cauchy loss (Li et al., 2018) (α = 0), Geman-McClure loss (Ganan & McClure, 1985), (α = −2), and Welsch (Dennis Jr & Welsch, 1978) loss (alpha = −∞). We also note that the L2 and Cross Entropy losses are some of the most commonly used losses and are often used without scale parameters throughout the deep learning community. We hope these comments speak to your concerns.
>
> > I would suggest that the authors should at least provide several examples by pointing out which is which.
>
> This is a great suggestion and we have added examples for each condition and dimensionality discussed and have highlighted these in dark yellow. Thanks!
>
> >In my opinion, the presentation of the paper could be further improved. For instance, the methodologies that the authors proposed become clear to me only after section 5.3. In addition, the title of this paper seems to be inappropriate in describing the main content of the paper.
>
> Thanks for this feedback, we agree that the first draft might have seemed a bit spread thin. We have tried to remedy this by simplifying and reworking a large portion of the paper (highlighted in light yellow), dropping some less impactful experiments, and adding significantly more experimental details. Also regarding “for instance, the methodologies that the authors proposed become clear to me only after section 5.3. In addition, the title of this paper seems to be inappropriate in describing the main content of the paper.” We are interested in understanding what we could do to signal our approach better. Is there a particular idea or statement that you think should be brought up earlier in the work.
>
> Regarding the title, we would be open to considering other options if you feel something could describe the main idea better. Please feel free to provide any additional feedback as we want to make this work as accessible as possible
>
> Again we thank you for the thoughtful feedback and hope that our changes can improve your outlook on the work. Please let us know if you think these changes address your feedback, or if you have any additional feedback that we can use to help improve the work.

---

### Official Review · AnonReviewer2 · 2020-11-01
**Work seems very interesting, but paper glosses over too many important points**

**Rating:** 4
**Confidence:** 4

**Review:**

This paper's starting point is a probabilistic interpretation of three losses: MSE, cross-entropy, and the loss introduced in reference (Barron 2019). It proposes the minimize the average loss jointly over the predictor's parameters $\theta$, but also the loss' own parameters $\phi$, vary, yielding an adaptive loss. It categorizes ways to define losses according to whether they depend on the data value, on the data index but not its value, or are independent of the data. The jointly optimized loss yields superior performance as re-calibrators (table 2 column CAL for regressors and table 5 for classifiers). It allows a natural definition of outlier detectors (sec  5.2)

The paper looks correct, specifically sec1 the approach of interpreting losses as NLL, sec4 the categorization, the experiment setup (though the description in the paper is cursory), the experiment design and use of CAL and ECE in sec5.4 on recalibration. The competitor methods and benchmarks for recalibration and outlier detection are quite exhaustive. (I am familiar with the references in section 3 related work)

The claims of superiority made in sec5.1, 5.2, 5.4 seem supported by the experimental evidence.

Despite these strengths, I had a hard time evaluating and understanding the experiments, and partly the concrete details of the mathematical setup. For experiments, this happened because the paper glosses over almost every aspect: design, task, model, training, to only show final results. (I did not consult the supplementary material.)  For instance fig 2 and 3 are cryptic (in the caption of fig 2, I guess "model" should be replaced by "predicted"? and fig 3 relates to either predicted or data conditionings?). Table 1 is not referenced or explained in the text; one can assume it relates to the CelebA experiment from sec 5.1, but what is the form of the data-dependent loss for the "predicted" case? Table 2, why are we speaking of temperature (which only is a parameter for softmax loss in classification tasks) in the case of this regression experiment? Sec5.2, there is no experimental detail of the SVHN (and results are not discussed) and ODDS experiment.

The paper should make connections to maximum likelihood and maximum a posteriori optimization more apparent. I'm further curious about connections to Bayes' risk (eg cf Minka 2001 "ERM is an incomplete inductive principle") and KL/Bregman divergences (eg Buja, Stuetzle, Shen 2005). What is meant by "optimizing a model's prior distribution" sec5.3 ? Does sec5.3 amount to MAP optimization with some suitably defined prior? Other losses (eg hinge, Huber loss) than the three mentioned here can be given probabilistic interpretations, what motivates their choice (especially the choice of the recent, hence little used, robust loss $\rho$) ?

I found it difficult to settle on a rating for this paper. The work behind it definitely seems to have strenghts, for instance, it looks interesting, and I might want to point it out to colleagues, it investigates an interesting, not yet mainstream probabilistic interpretation in a relatively simple way, with few ad hoc assumptions. Yet the paper on its own is lacking in detail so much that the experiments are impossible to understand fully
As a conference paper, it seems to be "bursting at the seams"; maybe it would be better suited as a journal paper, where the important aspects can be developed. According to ICLR guidelines, it could be made up to 10 pages long; sec2 could be made terser.

---

> ### Author Response · Authors · 2020-11-19
> **Response**
>
> We thank the reviewer for their detailed and thoughtful feedback! Our point-by-point reply follows, and a revision has been uploaded. Highlights for revisions (light yellow) and additions (dark yellow) are included.
>
> > "the paper glosses over almost every aspect: design, task, model, training, to only show final results."
>
> We appreciate the feedback to focus on fewer experiments and fuller details. To this end, we have removed the robust regression content and used the space to better cover outlier detection and add experimental details for all sections. We have likewise combined robust modelling and outlier detection into a single section. We unify these two because robustness and outlier detection are two sides of the same coin: for a model to be robust it must identify and “down-weight” outliers. In this light, we highlight CelebA VAE and ODDS outlier detection results, both of which apply likelihood parameters to unsupervised models.
>
>
> > "(I did not consult the supplementary material.)"
>
> We have incorporated the supplementary content in Sections I and J on experimental details into the text for easier reading.
>
> > "what is your data-dependent loss for the "predicted" case"
>
> The data dependant loss has the same basic form as  the other experimental settings (robust regression loss $\rho(x, \sigma, \alpha)$ except that the $\sigma$ and $\alpha$ are now predicted by the model. This allows the shape and scale of the loss to vary with each datapoint according to the learned model. (Note this contrasts with data parameters, which assign distinct likelihood parameters to each datapoint nonparametrically.) We have revised the text to clarify this.
>
> > "What is meant by "optimizing a model's prior distribution" sec5.3?"
>
> In addition to optimizing the shape and scale of the likelihood distribution of the model output, we can use the same approach to optimize the prior distribution of the model parameters. We have revised the text with this phrasing. Sec. 5.3 is indeed about MAP optimization where the prior distribution parameters (Gaussian variances, Laplace scales) are jointly optimized. In particular this is equivalent to MAP inference of a hierarchical prior and we have included this in our revision.
>
> > "The paper should make connections to maximum likelihood and maximum a posteriori optimization more apparent. I'm further curious about connections to Bayes' risk (eg cf Minka 2001 "ERM is an incomplete inductive principle") and KL/Bregman divergences (eg Buja, Stuetzle, Shen 2005)"
>
> We added a note on this in the adaptive prior section. Furthermore because this approach reduces to MAP optimization the connections to ERM and KL divergence minimization are exactly those that relate this approach to MAP inference.
>
> > "Other losses (eg hinge, Huber loss) [...] can be given probabilistic interpretations, what motivates their choice?"
>
> We chose softmax, squared error, and the robust loss rho because the first two are ubiquitous and the third has recently shown strong results for a variety of tasks. The third loss also generalizes many other losses used throughout the literature. Our VAE experiments (Table 1) show that there is further advantage to experimenting with different forms of the likelihood parameters (the rho loss paper only tried global parameters).
> We added sections in the background on why we chose these distributions. In particular we are interested in the robust loss because it naturally generalizes many common robust loss functions  with only a single extra parameter. This robust loss, ρ, has the interesting property that it generalizes several different loss functions commonly used in robust learning such as the L2 loss (α = 2), pseudo-huber loss (Charbonnier et al., 1997)(α = 1), Cauchy loss (Li et al., 2018) (α = 0), Geman-McClure loss (Ganan & McClure, 1985), (α = −2), and Welsch (Dennis Jr & Welsch, 1978) loss (alpha = −∞). It also serves as an example of a setup where we can optimize over shape and not just scale.
>
> > "it seems to be "bursting at the seams""..."it could be made up to 10 pages long"
>
> Please note the page limit was reduced to 8 this year—for rebuttal this has been raised to 9 and we have made use of this in the uploaded revision. We have reduced the number of experimental settings and explained these settings in greater detail in order to make this feel more complete.
>
>
> > fig 2 and 3 are cryptic
> > Table 1 is not referenced
> > Table 2, why are we speaking of temperature
>
> Thank you for catching the Figure 2 typo and missing reference for Table 1. We have fixed both, added more explanatory text, and revised captions accordingly.
>
>
> Again we thank you for the thoughtful feedback and hope that our changes can improve your outlook on the work. Please let us know if you think these changes address your feedback, or if you have any additional feedback that we can use to help improve the work.

---

### Decision · Program_Chairs · 2021-01-07
**Final Decision**

**Decision:**

Reject

**Comment:**

The reviewers agree that this paper has some strengths to it, and some commented that the revision improved the manuscript, but the paper remained borderline with no strong champions in its favor. The reviews are encouraging and suggest that the paper is a bit tightly packed for the conference format, and perhaps because it is dense it is hard to the strength and scope of contributions, while other relationships such as to the Bayesian context could be explored more fully. Multiple reviewers find that a longer improved version would "shine" in a better suited journal.

The decision to reject is independent of the fact that the authors seem to have violated the anonymity rules in the revised version.